# Adversarial Fairness with Elastic Weight Consolidation

## Abstract

A central goal of algorithmic fairness is to develop a non-discriminatory approach to a protected group. We study methods to improve accuracy for the worst-group, primarily when data distribution is unevenly distributed. We propose a method to enhance both accuracy and fairness for the worst-group using regularization based on Elastic Weight Consolidation (EWC). We mitigate socially undesirable biases for binary classification tasks by applying adversarial models. To maintain the critical parameters for predicting the target attribute, we regularize the model using the Fisher information, referred to as EWC. We confirm that learning the task using the UCI Adult (Census), CelebA, and Waterbirds datasets yields a better trade-off between accuracy and fairness than in previous studies. The experimental results on table and image datasets show that our proposed method achieves better fairness improvements than the previous methods, maintaining accuracy under widely-used fairness criteria.

## 1 Introduction

Machine learning algorithms have been used in high-risk applications and increasingly require fairness, accountability, and transparency. The growing concern that machine learning models can falsely discriminate against minorities and other protected groups when used to make decisions has received considerable attention. Recent studies have shown that neural networks with many parameters are more difficult to generalize fairness than classification errors (Deng et al., 2023). Regardless of the imbalance in training data distribution, neural network models can easily overfit fairness goals during training (Wang et al., 2019). This is especially true when classes are imbalanced.

Unbalanced datasets across labels and demographic groups are a common problem in real data. Generalizations in this inherent imbalance data have been extensively studied (Sagawa et al., 2020; Liu et al., 2021; Nam et al., 2022; Idrissi et al., 2022; Kirichenko et al., 2023). These solutions compete for accuracy in the worst-group. When the label of the target and each data has an attribute, the combination of each class and attribute is called a group. The group with the worst accuracy in each group is called the worst-group. These papers is to train a classifier that maximizes the performance of the worst tests in the whole groups. However, in studies focusing on the worst-group, how to improve fairness generalization should be discussed more.

We propose a practical and straightforward method to improve fairness while maintaining model accuracy to solve the problem of fairness generalization of supervised classification tasks on unbalanced datasets. Recent studies (Kirichenko et al., 2023; Lee et al., 2023) show that the performance of spurious correlations and distribution shifts matches or even improves by only fine-tuning the last layer instead of updating all model parameters. Our approach learns all parameters, not just fine-tuning the last layer. We use Elastic Weight Consolidation (EWC) to preserve critical parameters for higher accuracy from the pre-training parameters and learn the critical parameters for higher fairness by adversarial debiasing (Kirkpatrick et al., 2017; Zhang et al., 2018).

We perform experiments to test the effectiveness and efficiency of our methods on multiple domains, such as table and image datasets, which have Gender-biased datasets used in various fairness studies. We experiment on a dataset with biased attributes for males and females, respectively. We demonstrate that our method

can better balance the trade-off between prediction accuracy and fairness than previous studies and achieves state-of-the-art performance on popular spurious correlation benchmarks compared to novel methods. Our contributions are as follows:

- To maintain accuracy with bias mitigation using the adversarial fairness method, a regularization approach based on EWC is proposed.
- The trade-off between accuracy and fairness is visualized to achieve a better balance with a smaller decrease in accuracy under fairness constraints.
- The effectiveness of the proposed method is presented in multiple domains through experiments conducted on several different datasets.

## 2 Related Work

Fairness in machine learning has been the focus of research in recent years. Researchers have been actively developing methods to mitigate bias in machine learning algorithms that are aware of social fairness. Several methods, including resampling, reweighting, and data augmentation, have been developed and deployed in practice (Ziang, 2003; He & Garcia, 2009; An et al., 2021). Fairness methods are mainly classified into those that aim to mitigate bias at the pre-processing (Feldman et al., 2015), in-processing (Zemel et al., 2013; Edwards & Storkey, 2016; Zafar et al., 2017; Donini et al., 2018; Madras et al., 2018; Martinez et al., 2020; Lahoti et al., 2020), and post-processing steps (Kim et al., 2019). Techniques that mitigate bias during training are helpful because they are powerful and do not require sensitive attributes reasoning. For example, Zhang et al. (2018), Adel et al. (2019) and Wang et al. (2019) developed an adversarial debiasing method to reduce the pseudo-correlations between target labels and sensitive attributions.

Various studies have shown that machine learning models often perform significantly worse in minority groups than in majority groups. Many methods to improve group robustness are based on the Distributed Robust Optimization (DRO) framework (Ben-Tal et al., 2013; Hu et al., 2018; Zhang et al., 2021). Group DRO which minimizes maximum loss across groups, is widely used to obtain high performance in worst-group (Sagawa et al., 2020). There are also techniques to deal with worst-group problems by devising sampling when creating mini-batches from learning datasets. SUBG subsamples all groups so they are the same size as the smallest group when doing normal learning (Idrissi et al., 2022). SUBG is a simple and robust baseline. These methods consider access to group information during learning. There have also been proposed methods considering only access to class information without group labels while learning. Just Train Twice (JTT) initially trains the model with fewer epochs (Liu et al., 2021). It is a method of fine-tuning with negative examples, assuming that the negative examples of the model with normal learning include samples of the worst group. Recently, there have been proposals to improve the worst-group accuracy by fine-tuning the last layer instead of updating the whole parameter in the model. Deep Feature Reweighting (DFR) is inspired by transfer learning and is a time and computational resource-efficient method that maximizes worst-group performance by fine-tuning pre-trained models (Kirichenko et al., 2023).

Although several approaches have been proposed, there has not been much discussion on improving fairness generalization in studies focusing on worst-group. In this study, we propose a method for improving fairness and maximizing the accuracy of worst-group when the group distribution of training data is unbalanced.

## 3 Setup

### 3.1 Worst-group notation

When predicting the target label $y \in Y$ from the input $x \in X$, the standard learning goal is to find a model $\theta$ that minimizes the empirical risk (Vapnik, 1995).

$$\theta_{ERM} = \arg\min_{\theta \in \Theta} E_{(x,y) \sim P}[L_t(\theta; (\phi_t(x), y))]. \tag{1}$$

Here, $\Theta$ is a model family, $L_t$ is the loss function of target model $\phi_t$, and $P$ denotes the data-generating distribution.

Following on the previous work of Sagawa et al., we address the problem of minimizing worst-group losses (Sagawa et al., 2020). Groups are determined by target and attribute labels (such as sensitive and spurious correlated attributes). DRO is to train a parameterized model $\phi_t : X \to Y$ that minimizes worst-group expected losses on test samples. In the given dataset, each sample consists of an input $x \in X$, a target label $y \in Y$, and an attribute label $a \in A$. Let $L_t(y, \phi_t(x))$ is the loss function of this predictor $\phi_t$, then minimization of group risk in DRO is,

$$\theta_{DRO} = \arg\min_{\theta \in \Theta} \left\{ \max_{g \in G} E_{(x,y) \sim P_g}[L_t(\theta; (\phi_t(x), y))] \right\}. \tag{2}$$

Here, $P_g$ denotes the group-conditioned data-generating distribution, and $g$ is a group as an attribute pair $g := (y, a) \in Y \times A =: G$. SUBG uses a data-generating distribution where all groups are the same probability mass as the smallest group.

### 3.2 Fairness notation

This paper aims not only to improve the accuracy of worst-group, also to improve fairness. We focus on two standard fairness metrics: demographic parity and equalized odds. When a pre-trained model $\phi_t$ exists that predicts $Y$ from $X$, model $\phi_t$ is unfortunately biased owing to the pseudo-correlation between $A$ and $Y$. We aim to achieve one of the following fairness criteria:

$$\textbf{Demographic Parity (DP)} : \hat{Y} \perp A,$$
$$\textbf{Equalized Odds (EO)} : \hat{Y} \perp A|Y. \tag{3}$$

For binary classification tasks, the empirical fairness metrics are

$$\textbf{DP} : p(\hat{Y} = 1|A = 0) - p(\hat{Y} = 1|A = 1)$$
$$\textbf{EO} : p(\hat{Y} = 1|A = 0, Y = y) - p(\hat{Y} = 1|A = 1, Y = y), \tag{4}$$

where $A$ is the binary attribute label and $\hat{Y}$ is outputs of $\phi_t$. DP is computed as the difference between the rate of positive outcomes in unprivileged and privileged groups. EO is computed as the average absolute difference between the false positive rate and the true positive rate for the unprivileged and privileged groups. These indicators are zero when the model is perfectly fair.

## 4 Methods

We propose a practical and simple method to improve fairness while improving and maintaining model performance on worst-group. Our method utilizes regularization based on the knowledge of continuous learning and uses the adversarial debiasing method. We describe adversarial debiasing and a method that preserves critical parameters in continuous learning before the proposed method. Finally, we explain the proposed method that utilizes them. Finally, we explain the proposed method that utilizes them.

### 4.1 Adversarial Debiasing

Zhang et al. (2018) proposed adversarial debiasing (AD) to mitigate bias by training a target and an adversarial model simultaneously when undesired biases are included in the training data. The adversarial debiasing procedure takes inspiration from the GAN used to train a fair classifier. Using a GAN, the authors introduced a system of two neural networks through which the two neural networks compete with each other to achieve more fair predictions. Similarly, they built two models for adversarial debiasing. The first model is a classifier that predicts the target variable based on the input features (training data). The second model is an adversary that attempts to predict a sensitive attribute based on the predictions of the classifier model.

Figure 1(a) shows an overview of adversarial debiasing. Let $\phi_t$ be the target predictor trained to achieve the task of predicting $Y$ given $X$, and let $L_t(y, \hat{y})$ be the loss function of this predictor $\phi_t$. The output $\hat{y}$ of predictor $\phi_t$ is input into the adversarial model $\phi_a$. This adversarial model $\phi_a$ corresponds to the

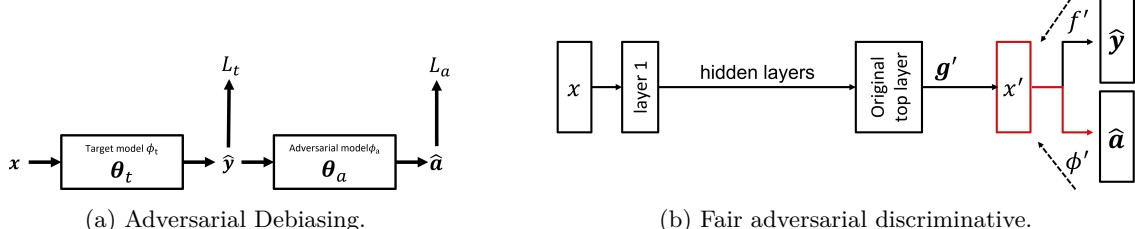

(a) Adversarial Debiasing.        (b) Fair adversarial discriminative.

Figure 1: Architecture of the adversarial debiasing and fair adversarial discriminative.

discriminator in the GAN. The adversarial model then attempts to predict the sensitive attribute values $a$ from the predicted $\hat{y}$ of the target model $\phi_t$. The adversarial model $\phi_a$ has a loss function $L_a(a, \hat{a})$, where $\hat{a}$ is the output of $\phi_a$, and $a$ is the sensitive attributes. Finally, the objectives that both models attempt to optimize are based on the predicted losses of the target and sensitive attributes, which are denoted by

$$L_{\text{total}} = L_t - \lambda_a L_a, \tag{5}$$

where $\lambda_a$ is a tuneable hyperparameter.

Adel et al. (2019) proposed an adversarial fairness method that can mitigate bias by slightly adjusting its architecture. Zhang et al. (2018)'s method requires a new model of the same size and is not memory-efficient when training a large model. The authors proposed a method called fair adversarial discriminative (FAD), in which a small discriminator is added to a shared middle layer of the original model.

An overview of the FAD is shown in Figure 1(b). Here, $g'$ is a new layer added to the original model to output a fair data representation $x'$. In addition, $f'$ is a classifier used to predict label $y$ of the original task from $x'$. Finally, $\phi'$ is a classifier that predicts sensitive attributes $a$ from $x'$. The dotted line represents the gradient, and the discriminator $\phi'$ is trained such that $x'$ acquires a fair feature representation through adversarial learning with a negative sign applied to the gradient.

### 4.2 Elastic Weight Consolidation (EWC)

Transfer learning, which uses models previously trained on large numbers of data, is a powerful technique to achieve high performance for a desired task (Yalniz et al., 2019). When a pre-trained model fits the distribution of the desired task through fine-tuning, it often forgets the original task distribution. This phenomenon is called catastrophic forgetting. Kirkpatrick et al. (2017) explored weighted regularization for continuous learning. They proposed EWC, a penalty applied to the difference between the parameters of the previous and new tasks. They regularized the current parameters to bring them closer to the parameters of the previous task. To retain those parameters that performed well in previous tasks during fine-tuning, they regularized the more influential parameters using Fisher information to avoid unnecessary updates. They used the diagonal components of the Fisher information matrix of the previous task parameters as weights for the influential parameters.

### 4.3 Proposed Method

We propose a regularization method based on EWC using adversarial debiasing for the classification problems with equation (3). Our method is based on the fine-tuning of a pre-trained model as in Figure 2. In the proposed method, "new task" and "old task" in EWC correspond to fine-tuning and pre-training, respectively. In this paper, the dataset was divided for fine-tuning and pre-training.

Figure 2(a) shows the architecture of the proposed method applied to adversarial debiasing. Given pre-training data $D_p$, the conditional probability of a class in pre-training is represented by $\phi_p(Y|X; \boldsymbol{\theta}p)$, where $\boldsymbol{\theta}p$ denotes the set of model parameters. The parameters $\boldsymbol{\theta}_p$ are learned to maximize the log-likelihood.

$$L(D_p; \boldsymbol{\theta}_p) = \sum_{(y_i, x_i) \in D_p} \log \phi_p(y_i|x_i; \boldsymbol{\theta}_p). \tag{6}$$

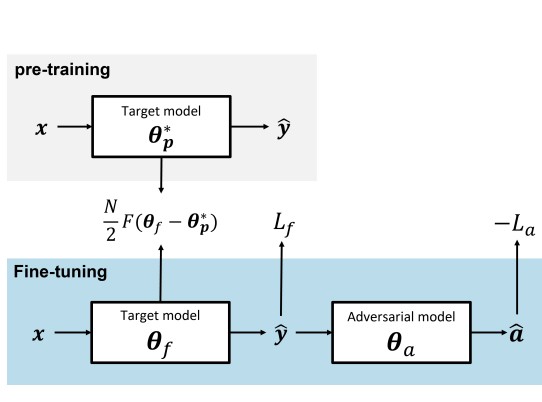

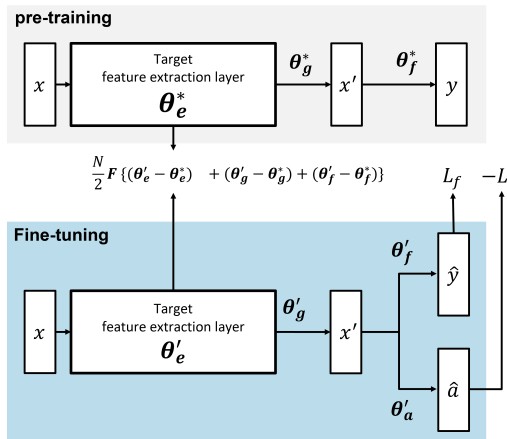

(a) Adversarial debiasing.

(b) Fair adversarial discriminative.

Figure 2: Architecture of the adversarial debiasing and fair adversarial discriminative.

Second, we consider bias mitigation using adversarial debiasing. Adversarial debiasing uses sensitive attributes $A$ in the adversarial model to suppress the social biases. When training a target model $\phi_f$ with adversarial model $\phi_a$ using fine-tuning data $D_f$,

$$L(D_f; \boldsymbol{\theta}_f, \boldsymbol{\theta}_a) = \sum_{(y_i, x_i, a_i) \in D_f} \left\{ \log \phi_f(y_i|x_i; \boldsymbol{\theta}_f) - \lambda_a \log \phi_a(a_i|\hat{y}_i; \boldsymbol{\theta}_a) \right\}, \tag{7}$$

where $\boldsymbol{\theta}_f$ is the training parameter of model $\phi_f$, with $\boldsymbol{\theta}_p$ as the initial value. Here, $\lambda_a$ is a tuneable hyperparameter, $\hat{y}_i$ is the output of $\phi_f(y_i|x_i; \boldsymbol{\theta}_f)$, and $a_i$ is a sensitive attribute label.

Next, we consider maintaining the accuracy using EWC during the fine-tuning process. Building on the concept of EWC, as discussed in the previous section 4.2, our approach utilizes Fisher information matrix $\boldsymbol{F}$ to regularize more influential parameters during fine-tuning, allowing us to preserve parameters that obtained good performance on pretraining and avoid unnecessary updates. The Fisher information matrix $\boldsymbol{F}$ is computed only from $\boldsymbol{\theta}_p^*$, and only its diagonal components are used. Using EWC, the parameters $\boldsymbol{\theta}_f$ are regularized during the fine-tuning such that they do not differ significantly from the fixed pre-training parameters $\boldsymbol{\theta}_p^*$. Therefore, the total objective function of the proposed method for adversarial debiasing is as follows:

$$L(D_f; \boldsymbol{\theta}_f, \boldsymbol{\theta}_a) = L_f(D_f; \boldsymbol{\theta}_f) - \lambda_a L_a(D_f; \boldsymbol{\theta}_f, \boldsymbol{\theta}_a) + \frac{N}{2} \boldsymbol{F}(\boldsymbol{\theta}_f - \boldsymbol{\theta}_p^*)^2, \tag{8}$$

where $L_f(D_f; \boldsymbol{\theta}_f)$ and $L_a(D_f; \boldsymbol{\theta}_f, \boldsymbol{\theta}_a)$ are objective functions of $\phi_f$ and $\phi_a$, $\boldsymbol{F}$ is Fisher information matrix and, $N$ is the number of data, respectively.

Figure 2 shows an architecture of the proposed method applied to FAD. In the Figure 2(b), let $\boldsymbol{\theta}_e'$ be the parameters of the feature extraction layer. Define $\boldsymbol{\theta}_f'$ and $\boldsymbol{\theta}_g'$ be the parameters of the classification layer. $\boldsymbol{\theta}_e'$ and $\boldsymbol{\theta}_g'$ are shared with the adversarial network. Let be $\boldsymbol{\theta}_p^* = \boldsymbol{\theta}_e^* + \boldsymbol{\theta}_g^* + \boldsymbol{\theta}_f^*$ and $\boldsymbol{\theta}_f = \boldsymbol{\theta}_e' + \boldsymbol{\theta}_g' + \boldsymbol{\theta}_f'$ the fixed pre-training and fine-tuning parameters respectively, the objective function of the FAD is the same as in equation (8).

In adversarial debiasing, the convergence of the target and adversary models is essential, and the target model must perfectly fool the adversary model while maintaining the level of accuracy (Zhang et al., 2018). Our proposal is aimed at updating the parameters involved in bias mitigation through an adversarial model and regularizing the parameters that are critical for the level of accuracy by applying an EWC. Moreover, our method does not optimize the loss function for each group like GroupDRO. Our method can also optimize the distribution of mini-batches the same as SUBG.

Table 1: Class and group counts for three fairness and worst-group benchmarks. These datasets exhibit group imbalance. CelebA has the lowest small group ratio, Waterbirds has the largest group size of non-diagonal components. Adult has an unbalanced distribution of targets and an unbalanced distribution of sensitive attributes.

| Dataset | Target | Group Counts | |
|---------|--------|--------|--------|
| Adult | $\downarrow y \quad a \rightarrow$ | Female | Male |
| | <= 50K | 23015 | 12911 |
| | >50K | 6149 | 1056 |
| CelebA | | Female | Male |
| | Blond | 22880 | 1387 |
| | Not blond | 71629 | 66874 |
| Waterbirds | | Water | Land |
| | Land bird | 56 | 1057 |
| | Water bird | 3498 | 184 |

## 5 Experiments

### 5.1 Experimental Conditions

#### 5.1.1 Datasets

Details of the datasets are shown in Table 1. We consider the following three common fairness and worst-group datasets used for bias mitigation:

(1) **Adult dataset** from the UCI repository Dua & Graff (2017) is used for predicting annual income based on various features. The adult dataset contains demographic information from the 1994 U.S. Census. The target variable for this dataset is whether the annual income is $> \$50K$, and gender is used as the sensitive attribute. The dataset was divided into training, validation, and testing groups. The proportions were 60% for training and 20% for validation and testing. This dataset contains categorical and continuous features and does not apply "fnlwgt". We discretized ten non-sensitive features without "sex", "race" and "fnlwgt" for input. We used binary $A$ for "male" and "female" with "sex" applied as a sensitive attribute. There are $n = 43131$ training examples and 1056 in the smallest group.

(2) **CelebA** is a large facial attribute dataset containing 202599 celebrity images, each with 40 attribute annotations Liu et al. (2015). CelebA has rich annotations for facial attributes. We used "Blond Hair" as the target attribute and gender ("Male") as the sensitive attribute $A$. The training, validation, and test percentages were split in the same proportions as the original paper, and we used the same data set as the original paper. There are $n = 162770$ training examples and 1387 in the smallest group.

(3) **Waterbirds** is a spurious correlation dataset that classifies water and land birds. This dataset was created by combining bird photos from the Caltech-UCSD Birds-200-2011 (CUB) dataset Wah et al. (2011) with image backgrounds from the Places dataset Zhou et al. (2018). The foreground is made up of $Y = \{\text{waterbirds}, \text{landbirds}\}$ and the background is made up of $A = \{\text{Waterbackground}, \text{landbackground}\}$ so that land birds appear more frequently against the water background. There are $n = 4795$ training examples and 56 in the smallest group.

#### 5.1.2 Baselines

We consider several popular worst-group accuracy methods and adversarial fairness. Empirical risk minimization (ERM) is the usual learning method that considers neither worst-group nor fairness. Group-DRO is a widely used method of adaptively weighting worst-group during training. SUBG is an ERM applied to a random subset of data where groups are evenly represented, and it is a simple yet powerful technique. Just Train Twice (JTT) is a method of detecting the worst-group by using only the group label of validation

data, not the group label of learning data. Finally, Deep Feature Reweighting (DFR) is a state-of-the-art method of fine-tuning the last layer of a pre-trained model with group weights for CelebA.

### 5.1.3  Models

For income classification within the Adult dataset, we used the multilayer perceptron (MLP) model, which includes the input, output, and three hidden layers. The hidden layers have 32 units and a rectified linear unit (ReLU) activation layer. During the training phase, each layer experiences a dropout with a ratio of 0.5.

For CelebA and Waterbirds, we used the standard ResNet-50, which was pre-trained on ImageNet and replaced the classification layers. We trained an adversarial model based on the FAD model Adel et al. (2019). We branched from the feature extraction layer of ResNet-50 into the target and sensitive classification layers.

## 5.2  Experimental Results

We experimented with bias mitigation using the three datasets compared with previous studies.

### 5.2.1  UCI Adult

The results for the adult dataset, which is generally applied in fairness experiments, are shown in Figure 3. The y-and x-axes represent worst-group accuracy and fairness metrics (DP and EO), respectively. The fair model has a DP and EO closer to 0. EO is a more difficult criterion than DP because the EO concept should correctly identify the positive outcome at equal rates across groups. Each scatter color represents different methods, and error bars represent Standard Deviation (STD). The dot scatter represents the proposed method, and the star scatter represents the previous studies. The each scatter was selected at the highest worst-group accuracy. In the proposed method, the dots represent the results for each hyperparameter $\lambda_a$ of the adversarial model. We can see that the difference in hyperparameters $\lambda_a$ changes accuracy and fairness. In the method using SUBG, the output was stable, but there was little change in fairness regardless of the adversarial parameters $\lambda_a$. Group DRO had the best accuracy, and SUBG had the best fairness value in previous methods. Figure 3(a) shows that the proposed method can get a better trade-off in fairness and worst-group accuracy than other related approaches. When SUBG is applied to the proposed method, it converges around some fairness values regardless of adversarial parameters but has the highest accuracy.

### 5.2.2  CelebA

Next, we experimented with the bias mitigation using CelebA. The results of which are shown in Figure 4. The y- and x-axes represent worst-group accuracy and fairness metrics (DP and EO). The fair model has a DP and EO closer to 0. Each scatter color represents different methods, and error bars represent STD. The dot scatter represents the proposed method, and the star scatter represents the previous studies. The each scatter was selected at the highest worst-group accuracy. In the proposed method, the dots represent the results for each hyperparameter $\lambda_a$ of the adversarial model. Figure 4 shows that the proposed method greatly improves previous methods' fairness and worst-group accuracy. In the case of CelebA, there was not much change when SUBG was applied to the proposed method. DFR had the best accuracy, and SUBG had the best fairness value in previous methods. A comparison of the highest worst-group accuracy with the average accuracy between the previous and the proposed methods is shown in Table 2. The proposed method shows the result of the parameter with the highest worst-group accuracy. Table 2 shows that the proposed method has the slightest difference between average and worst-group accuracy. The proposed method can get the best fairness because adversarial models attempt to equalize the accuracy of the whole group.

### 5.2.3  Waterbirds

Finally, we show the results of the Waterbirds dataset. The results of which are shown in Figure 5. The y-and x-axes represent worst-group accuracy and fairness metrics (DP and EO), respectively. The fair model has a DP and EO closer to 0. Each scatter color represents different methods, and error bars represent STD.

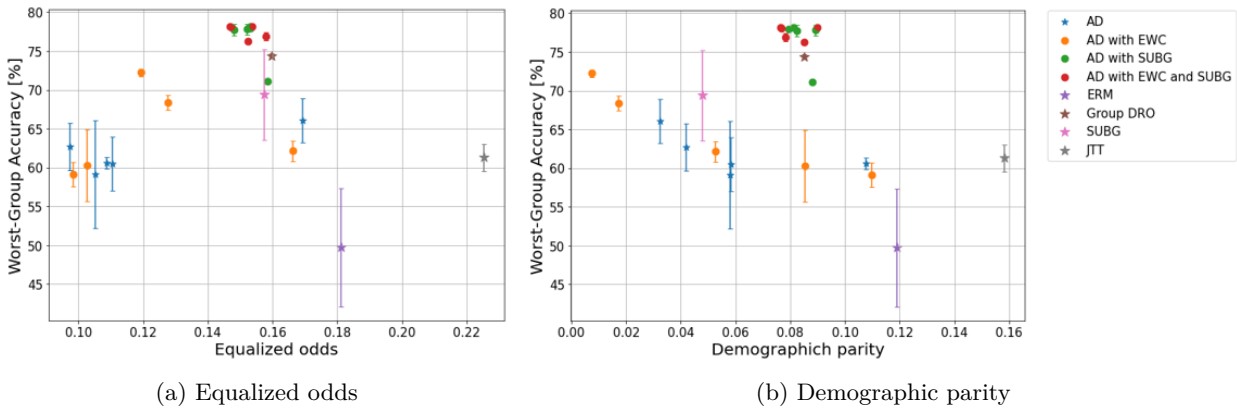

(a) Equalized odds        (b) Demographic parity

Figure 3: Results on the Adult dataset, left and right figures show EO and DP scores, respectively. We vary the tunable parameter $\lambda_a$ for our proposed methods ("AD with EWC", "AD with SUBG", and "AD with EWC and SUBG") to record the performance. The model is better as the scatters go to the upper left representing fairer and higher performance. We consider five different $\lambda_a = [0.5, 1, 10, 50, 100]$ for each proposed method. In the case of EO, accuracy and fairness are inversely proportional as the parameters $\lambda_a$ change. The proposed method achieves the best accuracy and fairness in both indices, showing a trade-off between good accuracy and fairness.

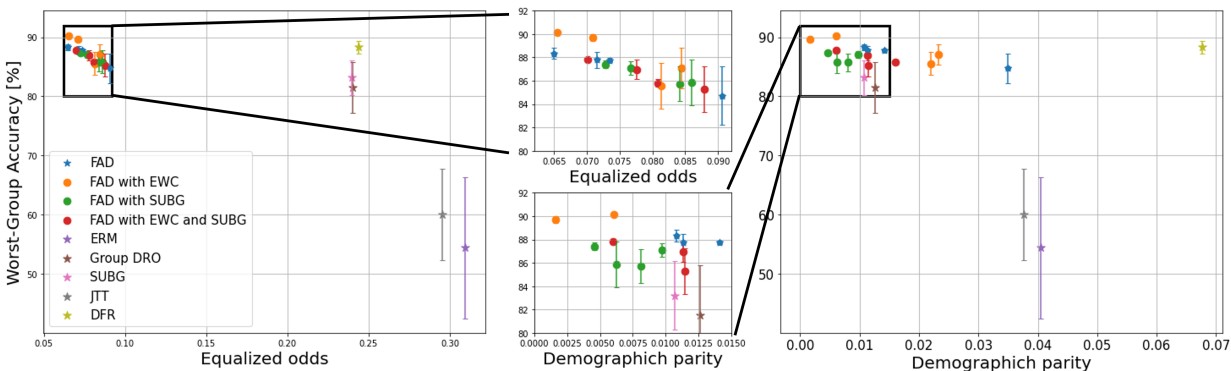

Figure 4: Results on CelebA dataset, left and right figure show EO and DP scores, respectively. We vary the tunable parameter $\lambda_a$ for our proposed methods ("AD with EWC", "AD with SUBG", and "AD with EWC and SUBG") to record the performance. The figure in the center shows an expanded area of the proposed method for each fairness measure. The model is better as the scatters go to the upper left representing fairer and higher performance. We consider four different $\lambda_a = [0.5, 1, 10, 100]$ for each proposed method. The proposed method achieves the best accuracy and fairness in both indices, showing a trade-off between good accuracy and fairness.

The dot scatter represents the proposed method, and the star scatter represents the previous studies. The each scatter was selected at the highest worst-group accuracy. In the proposed method, the dots represent the results for each hyperparameter $\lambda_a$ of the adversarial model. Figure 5 shows that SUBG had the best accuracy and fairness value in all methods. The proposed method achieves the best accuracy compared with previous methods except SUBG. The SUBG method has dramatically improved accuracy and fairness compared to other methods. SUBG has stable performance in Waterbirds, which is consistent with Sagawa et al. Idrissi et al. (2022). A comparison of the highest worst-group accuracy with the average accuracy between the previous and the proposed methods is shown in Table 3. The proposed method shows a good trade-off between accuracy and fairness.

Table 2: The highest worst-group and mean test accuracy of the proposed method and baselines on CelebA hair color prediction problems. For all methods, we report the mean $\pm$ standard deviation over five independent runs of the method. The proposed method (FAD with EWC from Figure 4) has the slightest difference between average and worst-group accuracy. It also had the lowest variance in worst-group accuracy.

| Methods | CelebA | |
| --- | --- | --- |
| | Mean(%) | Worst-goup(%) |
| ERM | 94.4 | 57.5 $\pm$ 11.9 |
| Group DRO | 93.5 | 81.4 $\pm$ 0.5 |
| SUBG | 92.1 | 83.2 $\pm$ 2.9 |
| JTT | 93.2 | 60.1 $\pm$ 7.5 |
| DFR | 91.3 | 88.3 $\pm$ 1.1 |
| Proposed Method | 90.7 | **90.2 $\pm$ 0.1** |

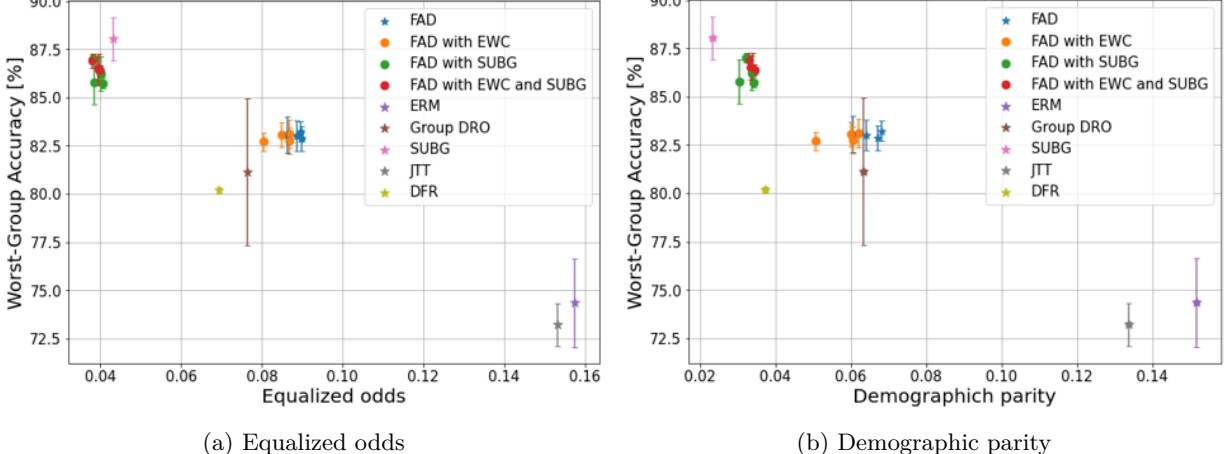

(a) Equalized odds         (b) Demographic parity

Figure 5: Results on Waterbirds dataset, left and right figure show EO and DP scores, respectively For our proposed methods ("AD with EWC", "AD with SUBG", and "AD with EWC and SUBG"), we vary the tunable parameter $\lambda_a$ to record the performance. The model is better as the scatters go to the upper left representing fairer and higher performance. We consider four different $\lambda_a = [0.5, 1, 10, 100]$ for each proposed method. The SUBG achieves the best accuracy and fairness in both indices.

Table 3: The highest worst-group and mean test accuracy of our proposed method and baselines on Waterbirds dataset. For all methods, we report the mean $\pm$ standard deviation over five independent runs of the method. The results of the proposed methods are the average values when the hyperparameters are changed. The proposed methods have good worst-group accuracy and low standard deviation.

| Methods | Waterbirds | |
| --- | --- | --- |
| | Mean(%) | Worst-goup(%) |
| ERM | 86.7 | 74.3 $\pm$ 2.3 |
| Group DRO | 90.2 | 81.1 $\pm$ 3.8 |
| SUBG | 89.2 | **88.1** $\pm$ 1.1 |
| JTT | 93.2 | 73.2 $\pm$ 1.1 |
| DFR | 91.3 | 80.2 $\pm$ **0.2** |
| FAD | 92.2 | 83.1 $\pm$ 0.7 |
| FAD with EWC | 92.5 | 82.9 $\pm$ 0.6 |
| FAD with SUBG | 92.8 | 86.3 $\pm$ 0.7 |
| FAD with EWC and SUBG | 92.7 | **86.6 $\pm$ 0.4** |

# 6 Conclusion

We have presented EWC-based regularization into the adversarial bias mitigation method. Adversarial bias mitigation methods have the issue of decreasing accuracy. In contrast, introducing EWC-based regularization helps maintain accuracy. We apply the method to two different domains: a table data classification task and an image attribution classification task. The resulting trade-off between fairness and accuracy was analyzed and evaluated. Our proposed method showed empirically higher accuracy than the related works and achieved a similar degree of bias mitigation. The results of applying our proposed method to table and image datasets also present that accuracy is maintained and bias is mitigated in multiple domains.

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
