# OpenReview forum: "Adversarial Fairness with Elastic Weight Consolidation"
_TMLR — Rejected by TMLR_

### Review · Reviewer_HuDV · 2023-09-21

**Summary Of Contributions:**

This paper proposes a training method to achieve good fairness while preserving accuracy for small groups within the data distribution where group distribution is highly imbalanced. The method combines fair adversarial training (to maintain fairness) with elastic weight consolidation (to limit accuracy degradation). Experiments on both tabular and image data show the proposed method can maintain a good fairness while preserving accuracy for the worst group.

**Audience:**

Yes

**Broader Impact Concerns:**

The submission can benefit from a border impact statement mentioning the limitations and study on fairness notions. The submission mainly focuses on DP and EO notions, and fairness can also be defined in other aspects. Just achieving fairness under DP and EO may not be read too much or be treated as achieving ultimate fairness.

**Claims And Evidence:**

No

**Requested Changes:**

1. Grammar error: page 1, paragraph 3, "is learned to simultaneously features ..."
2. What is the connection between "the performance of spurious correlations and distribution shifts matches or even improves by only fine-tuning the last layer instead of updating all model parameters" and "our method"?
3. Imprecise word: page 1, paragraph 4, "We demonstrate that our method can better examine the balance ..." It is not an evaluation method so "examine" is not appropriate.
4. Confusing words: page 2, contribution 2, "... during bias relaxation" - intuitively it should be "under fairness constraint"
5. Imprecise word - who are "they" in page 2, section 2, paragraph 1, "For example, they developed ..."
6. Not a sentence - page 3, first paragraph, "Where each sample of the dataset consists of an input ... "
7. Imprecise word - page 3, second paragraph, "... where all groups are the same size ..." - a distribution should have no size. It should be "probability mass".
8. Imprecise word - page 3, section 4, first paragraph, "... a method that maintains important parameter" - what does "maintain" mean here?
9. "equation equation" appears a few times, please fix.
10. Section 4.3, paragraph 2: definition of $\phi_p(Y|X;\theta_p)$ appears twice - redundant.
11. Equation (8): please explain $N$ and $F$, especially, given the loose space constriant, you may need to explain how $F$ is computed to be self-contained.
12. Section 4 last paragraph: "... when applying an EWC." -> "... by applying an EWC."
13. Missing "." in Page 7 Line 1 between "learning data" and "Finally".
14. Missing space in Conclusion: "regularizationhelps"
15. Not a sentence: Conclusion section "Demonstrates that in both ..."
16. What does "high performance" mean in "... our proposed method showed empirically higher performance than ..."? Is it higher fairness or higher accuracy.

**Strengths And Weaknesses:**

Strengths:
- An important and practical problem - the data distribution is usually highly imbalanced and fairness and accuracy are critical to maintain. The proposed method is an attractive solution for this scenario.
- The proposed method is reasonable and effective backed up by strong experimental evaluation. The idea of compiling fairness adversarial training and elastic weight consolidation is simple, natural, yet powerful and easy to deploy.

Disclaimer: I am not actively working in machine learning fairness, and I will read other in-domain reviewer's feedback to adjust my evaluation.

Weaknesses:
- The writing quality needs to be greatly improved. The current presentation is subpar to be published and readers need to spend much time guessing the actual mearning. There are several grammatical issues and imprecise word usage that hinder understanding. Due to this concern, I would recommend the authors revise the manuscript thoroughly and resubmit, since the manuscript does not give " accurate and clear" evidence. Some examples of writing issues are listed in "Requested Changes".

---

### Review · Reviewer_rRge · 2023-10-08

**Summary Of Contributions:**

The authors present a method for fairness in machine learning that tries to improve the accuracy and fairness in the worst-case group. The authors start from two previous fairness models, the Adversarial Debiasing (AD) [Zhang et al., 2018] and the Fair Adversarial Debiasing (FAD) [Adel et al., 2019]. However, rather than directly using those methods, the authors split the training tasks into pre-training and fine-tuning tasks using a continual learning framework, the Elastic Weight Consolidation (EWC) [Kirkpatrick et al., 2017]. In the pre-training task, the authors trained the model using a regular ERM model without considering fairness constraints. Then, the AD or FAD method is applied to the fine-tuning task, but they constrain the model's weights not to differ too much from the pre-trained weights via an EWC constraint. Additionally, the authors also incorporate SUBG, a subsampling technique that samples all groups with roughly the same size, to further improve the performance of the model in the worst-case group. Finally, the authors demonstrate the benefit of the models against baselines on three popular fair datasets.

**Audience:**

Yes

**Broader Impact Concerns:**

I do not have any broader impact concerns for this paper.

**Claims And Evidence:**

No

**Requested Changes:**

- Please fix some clarity and presentation issues I mentioned.
- Add additional baselines.
- Add more discussion on the experiment results.

**Strengths And Weaknesses:**

Strengths:
- Interesting exploration of using continual learning techniques for improving the fairness of the model while maintaining its accuracy, where the fairness constraints are only introduced during the fine-tuning task.
- The authors consider two base models, AD and FAD, in which the FAD is more suitable for larger networks.
- The results show that the proposed method is competitive over the baseline.

Weaknesses:
- Contributions. The proposed method is a rather direct incorporation of the Adversarial Debiasing model on top of a continual learning framework.
- Baselines. The worst-case group fairness is currently a popular fairness area, where many methods have been proposed [1-6]. I would expect the authors to include some of those methods as baselines in addition to the GroupDRO methods.
- Experiments. I would like to see more explanations in the experiment sections. For example: (1) why using AD only in the fine-tuning section perform better than fully using AD in the whole training; (2) how does the error bar for the ERM method calculated; (3) why the plot only have error bars for accuracy, but not fairness metric; (4) why the overall accuracy of the ERM model in Waterbird is lower than the overall accuracy of all fair models; etc.
- Presentation. In the method section, there are multiple subsections that talk about previous works (Sec 4.1 and Sec 4.2). I would suggest the authors focus on their contributions in the method section and move the discussion about previous works to Section 2. This will make it easier for the reader to check the main contribution of the paper.
- Clarity. I would suggest the authors improve the clarity of the paper by using writing assistance to polish the arguments and sentences. I sometimes have trouble understanding the main concept of several sentences.

Reference:
- [1] Martinez, Natalia, Martin Bertran, and Guillermo Sapiro. "Minimax pareto fairness: A multi objective perspective." International Conference on Machine Learning. PMLR, 2020.
- [2] Martinez, Natalia L., et al. "Blind pareto fairness and subgroup robustness." International Conference on Machine Learning. PMLR, 2021.
- [3] Diana, Emily, et al. "Minimax group fairness: Algorithms and experiments." Proceedings of the 2021 AAAI/ACM Conference on AI, Ethics, and Society. 2021.
- [4] Shekhar, Shubhanshu, et al. "Adaptive sampling for minimax fair classification." Advances in Neural Information Processing Systems 34 (2021): 24535-24544.
- [5] Abernethy, Jacob D., et al. "Active Sampling for Min-Max Fairness." International Conference on Machine Learning. PMLR, 2022.
- [6] Pethick, Thomas, Grigorios Chrysos, and Volkan Cevher. "Revisiting adversarial training for the worst-performing class." Transactions on Machine Learning Research (2022).

---

### Review · Reviewer_nLuA · 2023-10-09

**Summary Of Contributions:**

This paper focuses on improving the fairness (worst-group accuracy) while maintaining the overall accuracy. To this end, this paper proposes a method that introducing the Elastic Weight Consolidation (EWC) regularization to the adversarial debiasing paradigm. Experiments on multiple datasets demonstrates the effectiveness of the proposed method on improving the fairness while preserving mean accuracy.

**Audience:**

Yes

**Claims And Evidence:**

Yes

**Requested Changes:**

* Adding motivations *before* introducing the proposed method, making Section 4 read more naturally.
* Provide a name and corresponding abbreviation for the proposed method.
* Discuss the connection between this work and fairness in adversarial training.

**Strengths And Weaknesses:**

### Strengths
* This paper focuses on a popular and critical problem of machine learning.
* The proposed method is clear to understand. The equations, Figures 1 and 2 clear shows the paradigm of adversarial debiasing method. Equation (8) clearly shows the proposed loss function.
* The experiment settings are desirable. This paper conducted experiments on multiple popular datasets with various baselines involved. The authors also reported deviation on the results to ensure reproducibility.
* The experiment results shows the proposed method can notably improve fairness-accuracy trade-off.

### Weaknesses
* Though the proposed method is easily understood, the authors directly introduce the proposed method without illustrating motivation, making the motivation somewhat unclear (why use the EWC as the regularization for adversarial methods).
* The proposed method does not have a name, which is simply called "Proposed Method" and "FAD with EWC" in this paper.
* There are also researches [1, 2, 3] focusing on worst-class accuracy in adversarial training which is also referred to as "adversarial fairness". I wonder how adversarial debiasing connects to these methods.

[1] To be Robust or to be Fair: Towards Fairness in Adversarial Training, ICML 2023

[2] CFA: Class-wise Calibrated Fair Adversarial Training, CVPR 2023

[3] Hard Adversarial Example Mining for Improving Robust Fairness, arXiv 2308.01823

---

### Review · Reviewer_EtVY · 2023-10-15

**Summary Of Contributions:**

This paper proposes applying Elastic Weight Consolidation (EWC), a method for continual learning in neural networks, to improve the fairness of trained models while minimally compromising accuracy. EWC is embedded in two existing fairness methods: adversarial debiasing, and fair adversarial discriminative (FAD) models. The performance of the proposed methods is evaluated on three datasets that are frequently used in the fairness literature.

**Audience:**

Yes

**Claims And Evidence:**

No

**Requested Changes:**

Some specific phrases/sentences that I find unclear or misleading:
- "When the label of the target and each data has an attribute, the combination of each class and attribute is called a group."
- "the performance of the worst tests"
- "the performance of spurious correlations and distribution shifts matches or even improves"
- "Techniques that mitigate bias during training are helpful because they are powerful and do not require sensitive attributes reasoning."
- "DRO is to train a parameterized model $\phi_t : X \mapsto Y$ that minimizes worst-group
expected losses on test samples." This is true for the particular instance of DRO in Sagawa et al. (2020), but DRO is a much broader approach, so I find this misleading.
- "When a pre-trained model $\phi_t$ exists that predicts Y from X, model $\phi_t$ is unfortunately biased": This is certainly not always true.
- "the objectives that both models attempt to optimize are based on the predicted losses of the target and sensitive attributes": Only the Target model uses this loss, right? Doesn't the adversarial model only use the loss $L_a$?
- Figure 1(a) appears to only be accurate for DP. For EO, the adversary needs $(\widehat{y}, y)$, not just $\widehat{y}$.

Other issues:
- Change "table" to "tabular"
- "there has not been much discussion on improving fairness
generalization in studies focusing on worst-group." I disagree, and this seems to contradict the preceding part of the Related Work section.
- Equation (1) is not the empirical risk; it's the (unknown) population risk.
- Why are some things subscripted by t? What does the t refer to?
- It's confusing that the terms DP and EO are overloaded to refer to both the fairness criteria (Equation 3) and the disparities that represent departures from those criteria (Equation 4).
- Why is N/2 included in the loss in equation (8)? Overall, the paper would be benefit from additional intuition and explanation of those proposed methods.
- Typo in Figure 3(b): "Demographich"

**Strengths And Weaknesses:**

Strengths
---------
The idea of using EWC for fairness purposes is simple and has intuitive appeal. EWC aims to allow neural networks to learn new tasks without losing the ability to perform well on old tasks. In the present context, the "old task" corresponds to a pre-training phase in which a model is optimized for accuracy, and the "new task" corresponds to a fine-tuning phase in which the model is tuned for fairness, while an EWC-style penalty ensures that the tuned model lies close to the pre-trained model in parameter space.


Weaknesses
----------
- The experiment results are unclear. First of all, the "AD" and "FAD" baselines should be explicitly mentioned in the Baselines subsection (5.1.2). Second, why are the two methods applied on different datasets? Third, what do "AD with SUBG" and "FAD with SUBG" refer to? The two methods that are proposed and illustrated in Figure 2(a) and 2(b) both involve EWC, but the caption of Fig 3 refers to "AD with SUBG" as one of the authors' proposed methods.
- The descriptions of the results also are somewhat unclear. For example, the caption of Figure 3 says "In the case of EO, accuracy and fairness are inversely proportional," but I do not see this in the figure. I see the relationship between the values on the x and y axes varying non-monotonically for the proposed methods. That caption also says "The proposed method achieves the best accuracy and fairness in both indices, showing a trade-off between good accuracy and fairness." This doesn't appear to be true. For example, AD has the lowest disparity in 3(a).
- Outside the results, the language and notation are unclear in many places. For example, the notation in Equation (8) does not make sense. F is a matrix, the two parameters are vectors, and the loss is a scalar. I know what is intended in the last term of (8), but this notation does not express it. In some cases, I can't tell what the intended meaning of a sentence is. Specifics are given in the Requested Changes below.
- Given the overall lack of clarity, it's hard to confidently interpret the results or evaluate the claims of the paper.

---

### Decision · Action_Editor_Ps3b · 2023-11-12

**Recommendation:** Reject

**Comment:**

Four reviewers unanimously recommended rejection, owing to significant issues around clarity and discussion. From the AE's review these concerns are valid. We thus do not believe the submission to be suitable for publication at this time.

**Audience:**

The problem considered is of interest to the TMLR audience, and the ideas in the paper could be of interest as well if there are fixes to the clarity issues.

**Claims And Evidence:**

Reviewers found it difficult to assess the paper's central claims owing to a persistent lack of clarity in both mathematical notation and discussion of empirical results.

**Resubmission Of Major Revision:**

The authors may consider submitting a major revision at a later time.